# Trends of prescribing antimicrobial drugs for urinary tract infections in primary care in the Netherlands: a population-based cohort study

Marlies Mulder,[1,2] Esmé Baan,[3] Annelies Verbon,[4] Bruno Stricker,[1,2,5] Katia Verhamme[3,6]

For numbered affiliations see end of article.

**Correspondence to**
Dr Bruno Stricker;
b.stricker@erasmusmc.nl

## ABSTRACT

**Objective** Urinary tract infections (UTIs) are an important reason to consult a general practitioner (GP). Here, we describe antimicrobial drug prescribing patterns for UTIs by GPs in relation to the Dutch primary care guidelines.

**Methods** We conducted a population-based cohort study in the Dutch Integrated Primary Care Information (IPCI) database, which encompasses approximately 2.5 million patients. All patients aged ≥12 years with at least 1 year of follow-up from 1996 to 2014 were extracted from the database. The number of prescriptions and choice of drug type were investigated over time and in different age categories. The choice of antimicrobial drug classes for UTIs and the duration of nitrofurantoin use in women were compared with the Dutch primary care guidelines of 1989, 1999, 2005 and 2013.

**Results** The source population comprised 1 755 085 patients who received 2 019 335 antimicrobial drug prescriptions; 401 655 (35.1%) prescriptions were for UTIs (45.2% in women and 12.6% in men). The proportion of prescriptions for UTIs within all prescriptions with an indication code increased from 5.2% in 1996 to 14% in 2014 in men and from 28% in 1996 to 50% in 2014 in women. In men, UTIs were most frequently treated with fluoroquinolones during the entire study period, whereas fluoroquinolones were only advised as first choice in the latest guideline of 2013. In women, UTIs were increasingly (p<0.05) treated with nitrofuran derivatives with a statistically significant difference after implementation of the guideline of 2005. Compliance to the advised duration of nitrofurantoin prescriptions in women has increased since the guideline of 2005.

**Conclusions** Antimicrobial drug prescribing for UTIs seemed to have increased over time. Prescribing in line with the UTI guidelines increased with regard to choice and duration of antimicrobial drugs. We showed that databases like IPCI, in which prescription and indication are monitored, can be valuable antibiotic stewardship tools.

## INTRODUCTION

Urinary tract infections (UTIs) are among the most common infections in humans. UTIs cause a substantial burden of disease with

major economic consequences.[1 2] In women, UTIs are the most common reason to consult a general practitioner (GP) in The Netherlands with 232 contacts per 1000 women in 2014. In men, this number was substantially lower: 37 contacts per 1000 men.[3]

### Strengths and limitations of this study

► This study was conducted in a large database, including a substantial part (~15%) of the Dutch population, implying that the external validity of our study is high.
► Every individual in The Netherlands is assigned to a single general practitioner (GP), and antimicrobial drugs are not sold over the counter. This limits information bias, since we had access to all antimicrobial drug prescriptions prescribed in primary care.
► Improvement of coding discipline in more recent years might bias conclusions on total prescribing of antimicrobial drugs, but we assume that this bias would be similar for all indications. Furthermore, we compared the antimicrobial drug prescription data for urinary tract infections (UTIs) with antimicrobial drug prescription data for other infections.
► We analysed the choice of antimicrobial drugs prescribed by GPs in relation to the Dutch guidelines on treatment of UTIs and show that database studies can be used as a tool for good antimicrobial stewardship.

In most cases UTIs are treated with antimicrobial drugs. The choice of antimicrobial drug depends on the antimicrobial sensitivity of pathogens in urinary cultures (if taken), severity of the symptoms and potential comorbidities of the individual. Based on the resistance patterns of *Escherichia coli* in The Netherlands, Dutch guidelines on the treatment of UTIs have changed several times in the last few decades. The latest Dutch guideline for the treatment of UTIs in primary care was released in 2013 and recommends nitrofurantoin as first choice in all patients with cystitis,

**Table 1** Overview of the treatment of UTIs according to the Dutch guidelines for GPs

| | 1989 | 1999 | 2005 | 2013 |
|---|---|---|---|---|
| **Women** | | | | |
| *Cystitis* | Trimethoprim (J01EA) or sulfamethizol (J01EB) or nitrofurantoin (J01XE) 3 days | Nitrofurantoin (J01XE) 3 days or trimethoprim (J01EA) | First choice: nitrofurantoin (J01XE) 5 days  Second choice: trimethoprim (J01EA)  Third choice: fosfomycin (J01XX) | First choice: nitrofurantoin (J01XE) 5 days  Second choice: fosfomycin (J01XX)  Third choice: trimethoprim (J01EA) |
| *UTI with tissue invasion* | Amoxicillin (J01CA) | First choice: amoxicillin–clavulanic acid (J01CR)  Second choice: sulfamethoxazole–trimethoprim (J01EE) | First choice: amoxicillin–clavulanic acid (J01CR)  Second choice: sulfamethoxazole–trimethoprim (J01EE) or fluoroquinolone (J01MA) | First choice: ciprofloxacin (J01MA)  Second choice: amoxicillin–clavulanic acid (J01CR)  Third choice: sulfamethoxazole-trimethoprim (J01EE) |
| **Men** | | | | |
| *Cystitis* | UTI in men should always be considered as prostatitis | Nitrofurantoin (J01XE) *or* trimethoprim (J01EA) | First choice: nitrofurantoin (J01XE)  Second choice: trimethoprim (J01EA) | First choice: nitrofurantoin (J01XE)  Second choice: trimethoprim (J01EA) |
| *UTI with tissue invasion* | Trimethoprim (J01EA) *or* amoxicillin (J01CA) | First choice: Amoxicillin–clavulanic acid (J01CR)  Second choice: sulfamethoxazole–trimethoprim (J01EE) | First choice: amoxicillin–clavulanic acid (J01CR)  Second choice: sulfamethoxazole–trimethoprim (J01EE) or fluorochinolone (J01MA) | First choice: ciprofloxacin (J01MA)  Second choice: amoxicillin–clavulanic acid (J01CR)  Third choice: sulfamethoxazole–trimethoprim (J01EE) |

The table shows the treatment of UTIs according to the Dutch guidelines of 1989, 1999, 2005 and 2013. The recommended duration of treatment with nitrofurantoin is also described. Please note that information from guidelines on the treatment of pregnant women and also the treatment of risk groups, such as patients with diabetes or abnormalities of the urinary tract, differs and is not described. Furthermore, the treatment duration in men is, in all cases, longer than that in women.
GP, general practitioner; UTI, urinary tract infection.

including men. In case of signs of tissue invasion, ciprofloxacin is recommended as first choice.[4] Additionally, the recommended duration of treatment has changed. The recommended duration of treatment with nitrofurantoin for cystitis in women was 3 days in the guidelines of 1989 and 1999, whereas it has been extended to 5 days in the guidelines of 2005 and 2013 (table 1).[4–7]

Previous research shows that the use of antimicrobial drugs in the Netherlands in general has increased since 2005 until approximately 2012, most prominently in elderly patients, but has decreased again since then.[8 9] Since UTIs are a major reason to consult a GP, the use of antimicrobial drugs for UTIs contributes significantly to total antimicrobial consumption. Therefore, antibiotic stewardship focusing on (inappropriate) antimicrobial drug prescribing for UTIs is an important target in the fight against antimicrobial resistance (AMR). This study aims to study the choice of antimicrobial drugs prescribed for UTIs by GPs and the duration of nitrofurantoin use over time in relation to the Dutch national guidelines (of 1989, 1999, 2005 and 2013) for the treatment of UTIs in primary care using a large electronic primary care database.

## MATERIALS AND METHODS
### Data source
This study was conducted using data from the Integrated Primary Care Information (IPCI) database, which is a longitudinal observational dynamic database containing the records from more than 450 general practices throughout the Netherlands.[10] Briefly, IPCI contains the complete electronic medical records of ~2 500 000 patients, composed of, among others, data on diagnoses (coded and free text) and prescriptions coded according to the Anatomical Therapeutical Chemical (ATC) classification of the WHO.[11] The system complies with European Union guidelines on the use of data for medical research and has been proven valid for pharmacoepidemiological studies. More detailed information on IPCI has been described elsewhere.[10]

### Study population
The study cohort comprised patients aged ≥12 years with at least 1 year of valid database history in the IPCI database. The study period was from 1 January 1996 until 1 January 2015. Follow-up started from the following, whichever occurred last: start of the study period, age of 12 years or reaching a minimum of 12 months of database history. Follow-up ended when a patient left the database or died or at the end of the study period, whichever occurred first. Gender, age at the time of prescriptions) and follow-up time (time of each patient in the database) were assessed for all patients.

### Prescriptions
From the database, we selected all prescriptions for antimicrobial drugs prescribed during the study period,

using an automatic search on the ATC code 'J01', which is the ATC code for antimicrobial drugs. All prescriptions of antimicrobial drugs were further categorised by ATC drug class, for example, J01AA (tetracyclines) (supplementary table s1). We analysed all prescriptions of ATC class J01CA (penicillins with extended spectrum), J01CR (combinations of penicillins, including beta-lactamase inhibitors), J01EA (trimethoprim and derivatives), J01EE (combinations of sulfonamides and trimethoprim, including derivatives), J01MA (fluoroquinolones), J01XE (nitrofuran derivatives) and J01XX (other antimicrobials [mainly fosfomycin]). In addition, the duration of all prescriptions of nitrofurantoin (J01XE01) (which in the Netherlands is only prescribed for UTIs) was assessed.

### Indication of use of antimicrobial drugs

Indication of use was assessed through an automatic search on disease-specific codes. Antimicrobial drug prescriptions were linked to the indication of use through a unique patient identifier linking a prescription to a diagnosis using the International Classification of Primary Care (ICPC)-1 codes (version 5). These ICPC codes were categorised in the following indications: UTIs, skin infections, respiratory infections, ear infections or other infections (supplementary table s2). All prescriptions without an ICPC code were assigned to the group: 'no code for indication of use'. Urethritis was included in the group of 'other' infections and not in the UTI group because of the distinct pathophysiology.

### Analyses

The total number of prescriptions for UTIs, 'other infections' (including respiratory infections, skin infections and ear infections) and prescriptions without an indication code were calculated for the complete study time and per year. Additionally, the proportion of UTI prescriptions per calendar year was calculated with all antimicrobial drug prescriptions with an indication code as denominator. Also, all nitrofurantoin prescriptions were analysed per year.

Since both the number of users and the total number of antimicrobial drug prescriptions are interesting with regard to the study of AMR, we studied both. Because of the dynamic nature of the study cohort, the annual frequency of antimicrobial drug prescriptions was calculated by dividing the total number of antimicrobial drug prescriptions by the total number of person-years (PYs). The annual number of users per calendar year was calculated by dividing all users by the total number of PY in that specific year. With regard to the calculation of users, if an individual received prescriptions of more than one antimicrobial drug class in 1 year, the individual contributed data to the different classes. However, if an individual received two or more prescriptions of the same drug in 1 year, the individual contributed only once as a user of this specific drug class. The frequency of prescriptions and users were studied by age (12–17, 18–25, 26–35, 36–45, 46–55, 56–65, 66–75, 76–85 and ≥86 years age

categories), gender and calendar year. The prescribing of antimicrobial drugs by GPs was compared with the recommendations according to the national guidelines (table 1).[4–7]

We intended to investigate if improved coding over time has influenced the results of the frequency of prescribing antimicrobial drug prescriptions for UTIs in general, independent of the drug class. Therefore, we also studied the frequency of nitrofurantoin prescriptions over time (since nitrofurantoin was only prescribed for UTIs), and we studied the proportion of prescriptions for UTIs within total prescriptions with an indication code (thus including all antimicrobial drug prescriptions with information on indication of use).

Furthermore, all nitrofurantoin prescriptions prescribed to women during the study period were selected, and duration of use was categorised in 3, 5, 7 days or in 'other' (when the number of days of use was unknown or different from 3, 5, or 7 days). The proportion of each category with the total number of prescriptions of nitrofurantoin in women as denominator was calculated by year.

Finally, we used a time series ARIMA model (in SPSS V.24) in order to determine the effect of the implementation of the guideline of 2005.[12] The calendar year was first univariably added to the analysis to study differences of antimicrobial drug prescribing over time. Next, we ran a model including calendar year and the intervention (implementation of the guideline). When the p value of the interaction term was <0.05, the difference in slope before and after the implementation of the guideline was significant, implying a significant effect of the implementation of this guideline.

### Patients and public involvement

No patients were involved in any stage of this study.

### RESULTS

The study population comprised 1 755 085 patients aged ≥12 years with a mean follow-up time of 3.31 years. A total of 671 251 (38.2%) patients were prescribed at least one antimicrobial drug during the study period: 271 772 (40.5%) men and 399 479 (59.5%) women. In total, they received 2 019 335 antimicrobial drug prescriptions (mean of 3 prescriptions per person) during the study period. Of these prescriptions, 1 144 810 (56.7%) could be linked to an indication, namely, 528 464 (46.2%) antimicrobial drug prescriptions for respiratory infections; 401 655 (35.1%) for UTIs; 157 900 (13.8%) for skin infections; 29 984 (2.6%) for ear infections and 26 807 (2.3%) for 'other' infections.

### Antimicrobial drug prescriptions for UTIs by calendar year

For all years combined and only considering those prescriptions with indication of use, prescriptions for UTIs were 12.6% of the total prescriptions in men and 45.2% in women. The total number of prescriptions per year for

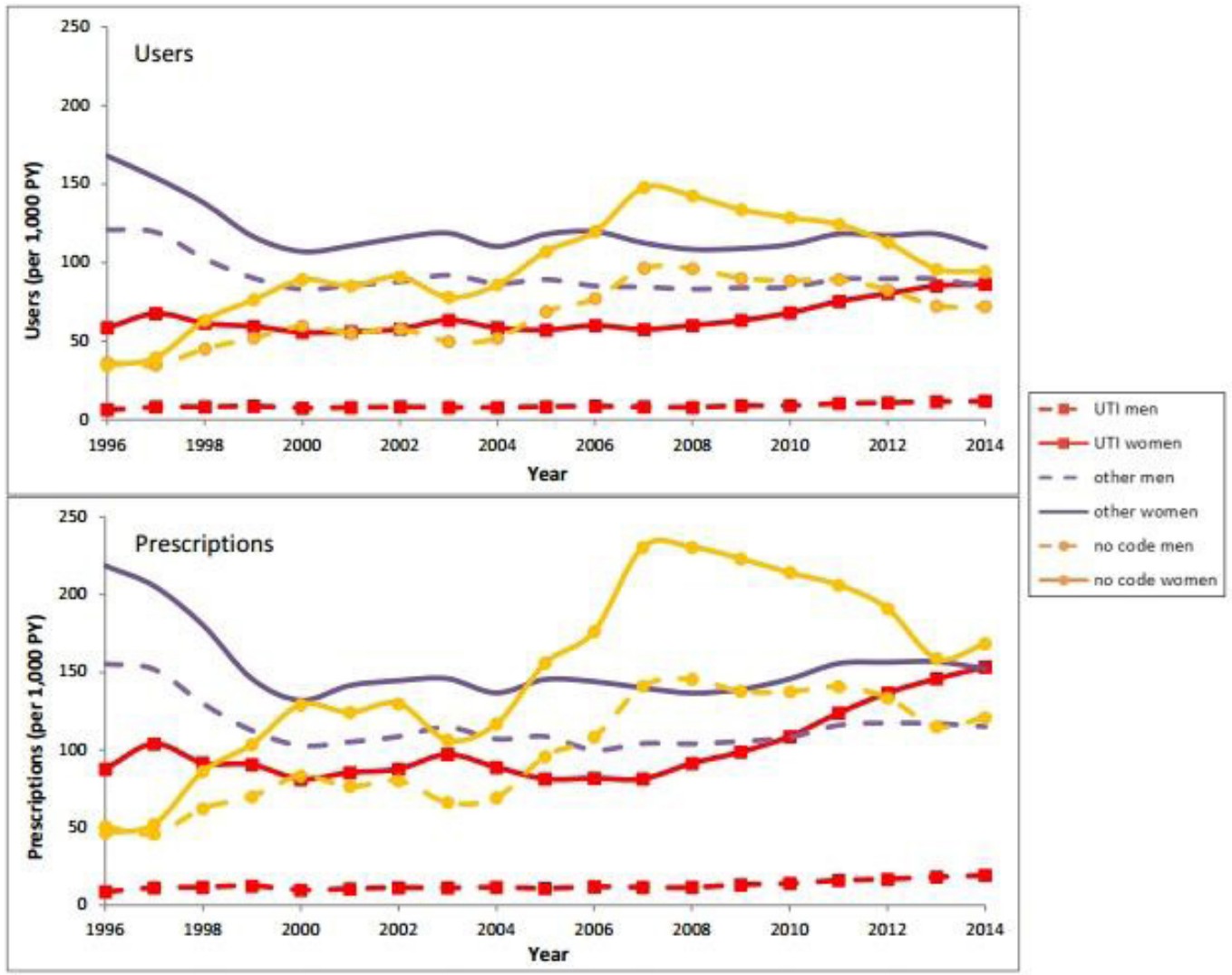

**Figure 1** The number of users and prescriptions of antimicrobial drugs for urinary tract infections, other infections and infections without indication code in the period 1996–2014.

UTIs increased from 9 (men) and 88 (women) prescriptions per 1000 PY in 1996 to 19 and 153 prescriptions per 1000 PY, respectively, in 2014. However, especially at the end of the study period, prescriptions without indication codes decreased (figure 1). To control for improvement of coding over time, we studied the total number of nitrofurantoin prescriptions, which in the Netherlands is only prescribed for UTIs. In women more clearly than in men, we see a flattening or even decrease of the number of non-coded nitrofurantoin prescriptions, whereas the number of coded and total nitrofurantoin prescriptions increased (supplementary figure s1). Moreover, we calculated the proportion of antimicrobial drug prescriptions for UTIs with all antimicrobial drug prescriptions with an indication code (for all indications) as denominator. In this analysis, the proportion of antimicrobial drugs for UTIs increased from 5.2% in 1996 to 14% in 2014 for men and from 28% in 1996 to 50% in 2014 for women (supplementary figure s2).

Fluoroquinolones (J01MA) are the most commonly prescribed antimicrobial drugs in men with UTIs with no clear increase or decrease over time. Other frequently prescribed antimicrobial drugs were combinations of sulfonamides and trimethoprim (J01EE) and combinations of penicillins, including beta-lactamase inhibitors (J01CR); the last group increased significantly over time until 2013. Also, in men, the number of prescriptions of nitrofuran derivatives increased significantly from 0.4 prescriptions per 1000 PY in 1996 to 6.2 prescriptions per 1000 PY in 2014. In women, nitrofuran derivatives (J01XE) were clearly the most frequently prescribed drugs since 1999, with a strong and significant increase in the last years from 52 prescriptions per 1000 PY in 2008 to 98 prescriptions per 1000 PY in 2014. They increasingly replaced the prescriptions of trimethoprim and derivatives (J01EA) and fluoroquinolones (J01MA), which were also commonly prescribed in women. Also, the prescriptions of combinations of penicillins, including

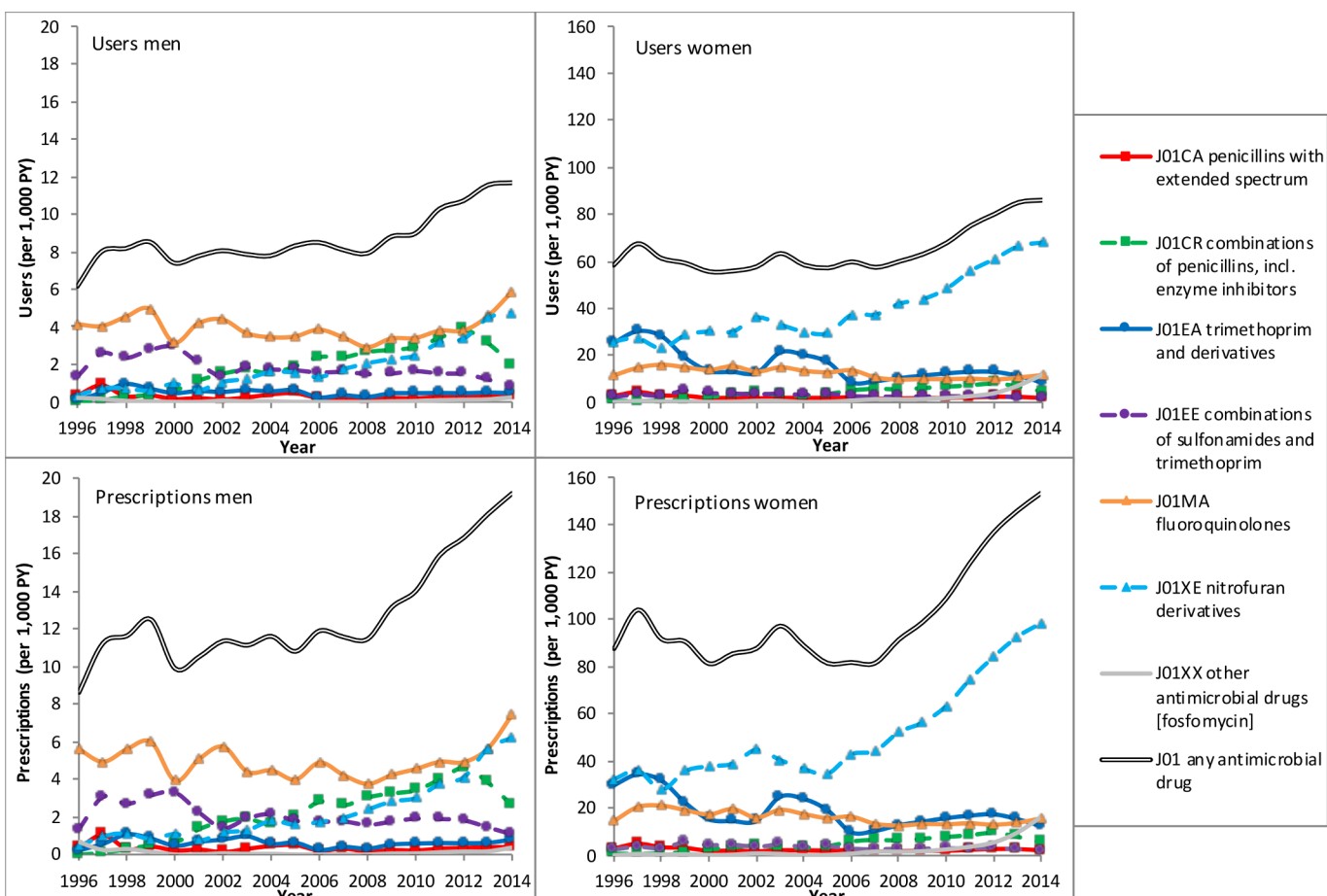

**Figure 2** The number of users and prescriptions of antimicrobial drugs for urinary tract infections in the period 1996–2014. Note that the scale of the y-axis differs between men and women.

beta-lactamase inhibitors (J01CR) and fosfomycin, significantly increased, for fosfomycin mainly in most recent years (figure 2).

### Choice of antimicrobial drug prescription in relation to the recommendations of the national primary care guidelines

The implementation of the Dutch guideline on the treatment of UTIs of 2005 was associated in women with a significant decrease in the slope of prescriptions of trimethoprim and derivatives (J01EA) and an increase in nitrofuran derivative (J01XE) prescriptions. Unfortunately, we did not have sufficient data before 1999 and after 2013 to study the implementation of the other guidelines. For men, no significant effects were found of the implementation of the guidelines on the slope of any antimicrobial drug prescriptions.

### Prescriptions of antimicrobial drugs for UTIs by age groups

Both in men and women, an increase in the total number of UTI prescriptions was observed for increasing age category. In women, the number of prescriptions in all age groups is higher than that in men, with nitrofuran derivatives (J01XE) as the most prescribed antimicrobial drug in all age groups, followed by fluoroquinolones (J01MA) and trimethoprim and derivatives (J01EA). In men, fluoroquinolones (J01MA) were the most frequently

prescribed antimicrobial drugs in most age groups, followed by nitrofuran derivatives (J01XE) and combinations of penicillins, including beta-lactamase inhibitors (J01CR) (figure 3).

Analysing the total number of prescriptions for UTIs (J01—antibacterials for systemic use) per age category, we observed an increase in the past years for all age categories, but most prominent in the higher age categories (supplementary figure s3).

### Changes in duration of nitrofurantoin prescriptions

In total, 215 531 prescriptions of nitrofurantoin with an ICPC code for UTIs were prescribed to women in the study period. In line with a change in recommended duration from 3 to 5 days in 2005, the proportion of prescriptions with a duration of 5 days increased strongly from 2005 onwards, while the proportion of prescriptions with a duration of 3 days became scarce. The proportion of prescriptions with a duration of 7 days decreased from 33% in 1996 to 18% in 2014. The proportion of prescriptions with a duration other than 3, 5 or 7 days (defined as 'other duration') remained stable over time, with a proportion ranging between 6% and 10% (figure 4).

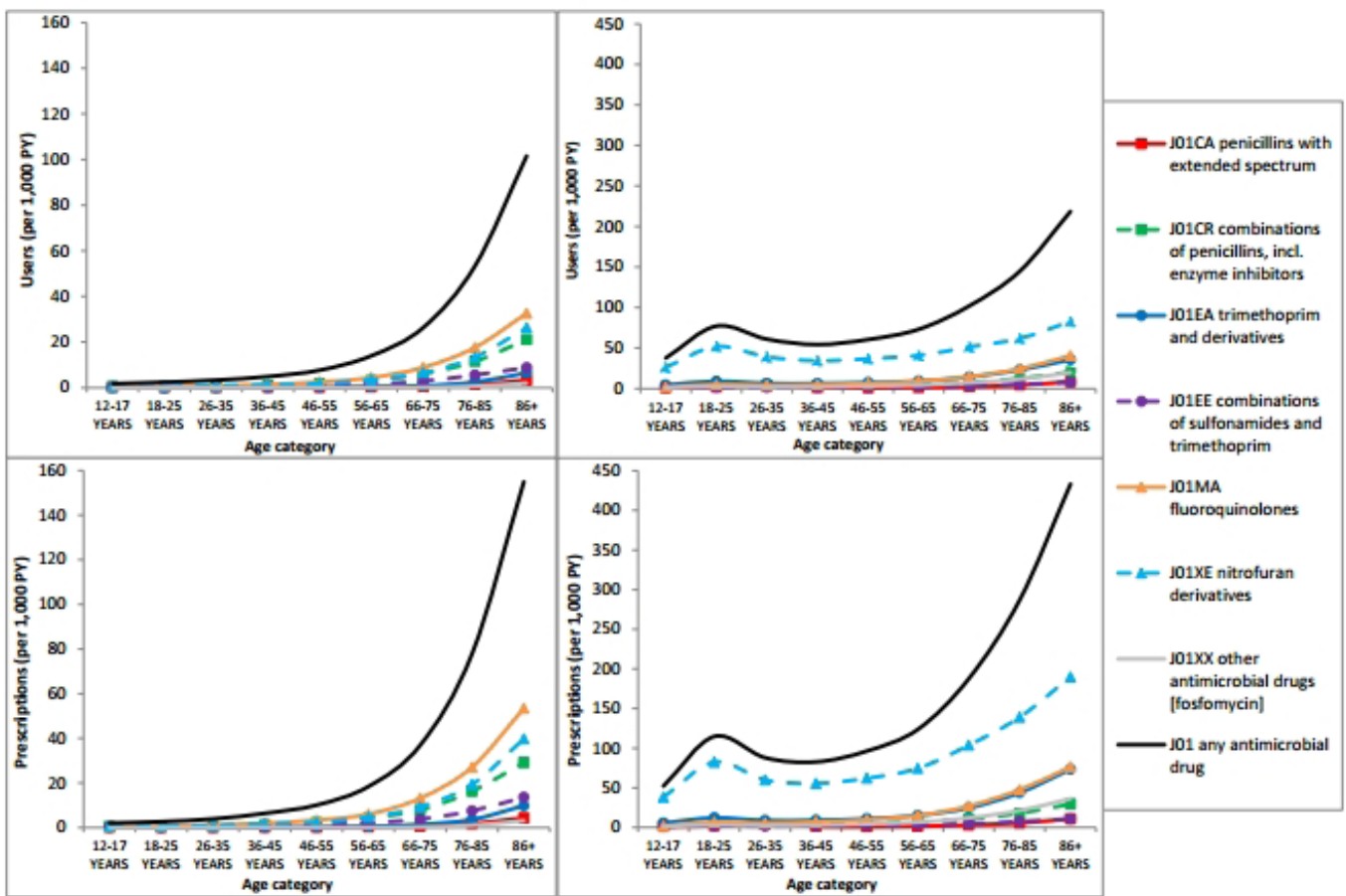

**Figure 3** The number of users and prescriptions of antimicrobial drugs for urinary tract infections per age group. Note that the scale of the y-axis differs between men and women.

## DISCUSSION

In this study, we investigated the prescribing of antimicrobial drugs for UTIs in primary care during the study period from 1 January 1996 to 1 January 2015 and compared this with the recommendations in the Dutch primary care guidelines. These guidelines are actively distributed by the Dutch College of General Practitioners.

During the study period, nitrofurantoin was shown to be the most prescribed drug for the treatment of UTIs, especially in women, and increasingly in men. The Dutch guidelines of 1989 indicated nitrofurantoin as one of the options for women, whereas it was suggested as option for both men and women in the guideline of 1999 and as the first-choice drug for cystitis in the 2005 and 2013 guideline. This is reflected in our findings; we observed an increase in nitrofurantoin prescriptions in men and women since 1999, with a steep and significant increase since 2005. In this period, nitrofurantoin replaced trimethoprim in women, which significantly decreased after 2005. Furthermore, in the guidelines of 1999 and 2005, amoxicillin–clavulanic acid was recommended for UTIs with tissue invasion, whereas amoxicillin (or trimethoprim in men) was recommended for UTIs with tissue invasion in the guideline prior to 1999. In men and women, a significant increase of combinations of

penicillins, including beta-lactamase inhibitors such as amoxicillin–clavulanic acid, was shown during the study period. In men, a flattening was seen from 2012, just before the guideline of 2013, which recommended ciprofloxacin as first choice.

We also studied the compliance of GPs to guidelines with regard to the duration of nitrofurantoin prescribing in women. The recommended duration increased from 3 to 5 days in 2005, which is reflected in our findings, indicating an increase in adherence to the guidelines by GPs, mainly since the guidelines of 2005 and 2013.[4–7] A deviation from the UTI guidelines was observed with regard to the prescribing of fluoroquinolones. Fluoroquinolones, in particular, ciprofloxacin, were only advised as first-choice drug for UTIs with tissue invasion from 2013 on (in 2005, it was second choice, and in 1989, its use was discouraged), whereas in men, these drugs were the most frequently described antimicrobial drug during the complete study period. In women, we saw a significant decrease over time, suggesting an increase of compliance of GPs over time.

A study in another Dutch GP database investigated antimicrobial prescribing for several infections during 2001 and showed that approximately 75% of the prescriptions for cystitis were first-choice drugs (nitrofurantoin or

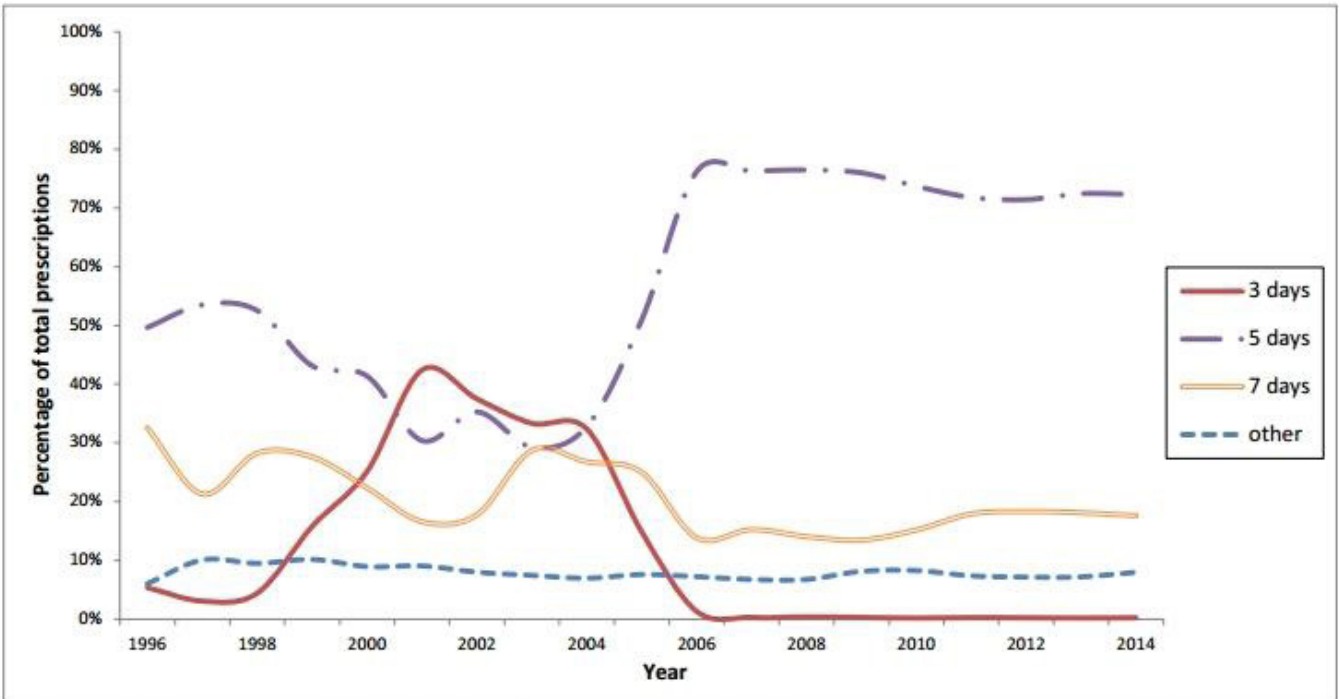

**Figure 4** The proportion of the different durations of nitrofurantoin prescriptions in women in the period 1996–2014. The proportion of prescriptions of nitrofurantoin (J01XXE01) with a duration of 3, 5 and 7 days or 'other' duration (unequal to 3, 5 and 7 days or unknown) with all these nitrofurantoin prescriptions as denominator. Until 2005, the duration recommended by the guideline was 3 days, whereas since the guideline of 2005, the advised duration was 5 days.

trimethoprim) according to guidelines.[13] The percentage of first-choice drug prescriptions in our study in 2001 is comparable (approximately 67% for women). A smaller study from 2009 in 970 non-pregnant women of ≥ 11 years with cystitis determined that 66% were prescribed nitrofurantoin, whereas we found a proportion of 57% in all women in 2009.[14] Our proportion is somewhat smaller, possibly because we did not distinguish between cystitis and pyelonephritis, since a proportion of the prescriptions had an indication code for both. However, the sample size of the other study is much smaller, resulting in more uncertainty around the true prevalence of antimicrobial drug use.

As part of new health quality standards, GPs are encouraged to improve disease coding, a phenomenon also observed in other countries. This might in part explain the increase in total prescriptions for UTIs, which is underlined by our finding that the number of prescriptions for nitrofurantoin without an indication code seemed to decrease, especially in women since approximately 2007. To control for this bias, we analysed UTI prescriptions as a proportion of the total number of antimicrobial drug prescriptions with an indication code (thus excluding prescriptions without a disease code). Yet, we observed that the increase of prescriptions was larger for UTIs than for other indications. This increase in antimicrobial drug prescribing for the treatment of UTIs is supported by other literature. According to the NIVEL (Netherlands Institute for Health Services Research) registration, the number of contacts with a GP for UTIs increased

(especially in women) from 185 per 1000 in 2012 to 295 per 1000 in 2015.[3] This was confirmed by another large Dutch GP database, showing an increase of GP visits for UTIs and antimicrobial drug prescriptions for UTIs in the period 2007–2010.[15 16] This increase of use of antimicrobial drugs for UTIs might partly be explained by ageing of the population. Indeed, we showed that antimicrobial drugs for UTIs were especially prescribed to patients in older age categories. Additionally, a Dutch study from 2012 also showed an increase of antimicrobial drug prescribing by age, and an increase of total prescriptions of antimicrobial drugs to elderly: in 2000, 9% of patients above 80 years had at least one prescription, which increased to 22% in 2009.[8] However, ageing of the population cannot solely explain the increase, since we also showed an increase in prescriptions of antimicrobial drugs in younger age groups. Possible explanations might be that more patients suffer from recurrent UTIs, that patients more frequently visit their GP in case of a UTI or that GPs more easily prescribe antimicrobial drugs for UTIs, but this has to be elucidated in future studies.

One of the strengths of this study is that we used a large population-based cohort from a database with detailed information on prescriptions and where information is collected as part of routine clinical care, reducing the risk of selection and information bias. We used these data to show that they can be of importance in studying antibiotic stewardship. Unfortunately, because of confidentiality, we could not study differences in prescribing patterns by region nor by GP characteristics such as age and/or

gender. Additionally, although we found that antimicrobial drugs were increasingly prescribed according to the guidelines, the study design did not allow us to investigate the reasons behind deviations from the guidelines. This is of interest and should be studied further, eventually by qualitative prospective research to distinguish between forced deviations caused by resistant pathogens and other deviations, which should be diminished to promote good antimicrobial stewardship.

Reducing the use of antimicrobial drugs is important, since the drawback of treating infections with antimicrobial drugs is the development of AMR.[17] Patients suffering from infections caused by resistant bacteria have an increased risk of a worse clinical outcome and death.[17] Moreover, AMR leads to an increase in healthcare costs, which has been estimated to be 1.5 billion euros in EU every year.[18 19] Misuse and overuse of antimicrobial drugs are the most important risk factors of AMR worldwide.[20] Therefore, antimicrobial stewardship programmes have been developed, aiming to optimise the prescribing of antimicrobial drugs and to minimise misuse and overuse in order to minimise AMR.[21] Our study demonstrates that databases with antimicrobial drug prescriptions and indications are useful for surveillance of antimicrobial drug prescriptions over time. In future, information gathered in such databases should be provided to GP practices as part of antibiotic stewardship in primary care.

**Author affiliations**
¹Department of Epidemiology, Erasmus Medical Centre, Rotterdam, The Netherlands
²Health and Youth Care Inspectorate, Heerlen, The Netherlands
³Department of Medical Informatics, Erasmus Medical Centre, Rotterdam, The Netherlands
⁴Department of Medical Microbiology and Infectious Diseases, Erasmus Medical Centre, Rotterdam, The Netherlands
⁵Department of Internal Medicine, Erasmus Medical Centre, Rotterdam, The Netherlands
⁶Department of Bioanalysis, Faculty of Pharmaceutical Sciences, University of Ghent, Ghent, Belgium

**Correction notice** This article has been corrected since it first published online. The open access licence type has been amended.

**Contributors** MM, AV, BS and KV designed the study. MM, EJB and KV analysed the data and interpreted these with the aid of AV and BS. MM wrote the manuscript with critical input from all coauthors.

**Funding** The authors have not declared a specific grant for this research from any funding agency in the public, commercial or not-for-profit sectors.

**Competing interests** KV works for a department that in the past received unconditional grants from Pfizer/BI, Yamanouchi, GSK and Novartis, none of which are related to the content of this paper. The other authors have nothing to declare.

**Patient consent for publication** Not required.

**Ethics approval** The scientific and ethical advisory board of IPCI approved this study (nr 11/2015).

**Provenance and peer review** Not commissioned; externally peer reviewed.

**Data sharing statement** The aggregated data are available with the corresponding author upon reasonable request.

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
