## [Reviewer comments · BMJ Open]

ARTICLE DETAILS

TITLE (PROVISIONAL)	Trends of prescribing antimicrobial drugs for urinary tract infections in primary care in the Netherlands: a population-based cohort study
AUTHORS	Mulder, Marlies; Baan, Esme; Verbon, Annelies; Stricker, Bruno; Verhamme, Katia

VERSION 1 - REVIEW

REVIEWER	SARA MALO University of Zaragoza, Spain
REVIEW RETURNED	24-Oct-2018

GENERAL COMMENTS	Monitoring antibiotic prescribing practice is an issue of special relevance for worldwide public health. The study reviewed analyses trends of antimicrobial prescribing for UTI in the Netherlands. However, I have doubts about if the second part of the objective “.. and investigate adherence of prescribers to Dutch primary care guidelines” is fully reached. To this end, it would be necessary to have access to additional information regarding the clinical condition of the patients included, the risks associated (pregnancy, diabetes, etc.) and/or if there existed a tissue invasion. Thus, I suggest that this aim is rewritten. Additionally, I would like to comment some other improvable points: Abstract: - explain the meaning of IPCI the first time it appears- clarify the difference between the two populations described (approximately 2.5 million patients and 1,755,085 patients)- indicate the guidelines versions used (in the results you mention the 2005 and 2013)- in the results: it is not clear if the increase in the proportion of prescriptions for UTI within all prescriptions with a code is due to a change in the number of prescriptions with a code or to a real increase in the number of prescriptions for UTIs. Moreover, be cautious in the interpretation of the results obtained (e.g. “with a significant effect of the guideline of 2005”) Strengths and limitations: please, explain better / improve the writing of the second and third points. They both result confusing Introduction: instead of “a large GP database”, please use a more accurate terminology Methods: - The subsection “Study population” refers, additionally, to the type of study, data sources and variables studied. Please, indicate it.
--

	- What does the inclusion criteria “with at least one year of valid database history” mean? Could it have any effect on the study population selection? - Which version of ICPC codes did you use? Detail the ICPC codes included in each indication group. - Analyses: from my point of view, presenting the results as number/proportion of prescriptions and users results confusing and difficult to understand. My recommendation is including just prescribing rates (proportion of users), as most studies do. - You mention the calculation of prevalence, but it is an incidence. Please, correct it. - The last paragraph in the Methodology section must be rewritten. It is not clear. Moreover, you refer to “calendar time” throughout the text. What do you exactly mean? Results: - I do not understand what the mean follow-up time means. Please clarify it according to the information provided in the Methods section. - Only 56.7% of the prescriptions were linked to an indication. Discuss the possible bias - Ideas described in lines 158-161 and 167-173 refer to Methodology and should be included in the corresponding section. - please explain the meaning of AB acronym - the title of the subsection “Adherence to guidelines” should be modified according to the idea explained at the beginning of this comment (the objective set is not completely reached, because you do not study adherence to guidelines. You are exploring the possible effect of new recommendations on the choice and duration of treatment, but you cannot attribute the changes in prescribing pattern to changes in guidelines) Discussion: first and second paragraphs contain repeated interpretations and ideas. Please, sum them up. Moreover, I consider that the results obtained should be discussed and interpreted more in depth. If it is lacking, then the manuscript results a bit simple and trivial since it consists on a simple descriptive analysis. Although more clinical data available would be needed in order to assess the adherence, an interesting discussion is behind these data so I encourage to improve its discussion. References: there are some errors in bibliography, such as the lack of uniformity in references or the incorrect organizations name. Figures: -please, simplify the figure legends. They contain ideas already mentioned in the text. - As I previously mentioned, it results confusing to include the users and prescription results - Modify the colors in order to facilitate interpretation when printed in black and white. Supplementary table 1 needs a more descriptive title.
--	---

REVIEWER	Alan Johnson National Infection Service Public Health England UK
REVIEW RETURNED	14-Jan-2019

GENERAL COMMENTS	In an era of increasing appreciation of the public health threat posed by antibiotic resistance and the need for antimicrobial
--

	stewardship, this study addresses an important and topical issue. The main conclusion highlighted by the authors is that databases with antimicrobial drug prescriptions and indications are useful for surveillance of antimicrobial drug prescribing over time. However, similar studies have previously been published in other countries that have established this principle and the main novelty of this study is thus restricted to the description of a surveillance system that can be used in the Netherlands. The study throws up some potentially interesting findings in terms of adherence of prescribers to national guidelines and the changes over time. However, at this stage the findings are primarily descriptive with little insight into the underlying causes. This severely limits both understanding of what factors influence the behaviour of prescribers and options for developing potential interventions to improve stewardship and the quality of prescribing at the present time. The authors are to be encouraged to continue their surveillance but further work is needed to give more insight into the factors influencing prescribing behaviour. For example, what is known of the variation in patterns of GP prescribing of antibiotics? Is there significant geographical/regional variation and if so does this correlate with other factors such as urban/rural settings or levels of deprivation. When new treatment guidelines are issued in the Netherlands, what is the process for disseminating this information to prescribers and might it result in variation in uptake by (e.g. are prescribers actively notified of changes or is the onus on them to try to keep up to date with guidance). Are data available as to the ages of GPs and how long they have been in practice? Might older GPs be more set in their ways and less amenable to changing their prescribing practice when new guidelines are issued? Additional information of this nature will serve to markedly improve the value and translational application of this potentially valuable work.
--	--

VERSION 1 – AUTHOR RESPONSE

Reviewer: 1

Monitoring antibiotic prescribing practice is an issue of special relevance for worldwide public health. The study reviewed analyses trends of antimicrobial prescribing for UTI in the Netherlands. However, I have doubts about if the second part of the objective “.. and investigate adherence of prescribers to Dutch primary care guidelines” is fully reached. To this end, it would be necessary to have access to additional information regarding the clinical condition of the patients included, the risks associated (pregnancy, diabetes, etc.) and/or if there existed a tissue invasion. Thus, I suggest that this aim is rewritten.

Thank you for this suggestion. We agree that the guidelines take pregnancy into account but unfortunately, unfortunately, not all pregnancies are registered in the GP practice. However, in the Netherlands most prescribing of antimicrobial drugs for UTI is relatively independent of co-morbidity. Furthermore, drugs such as nitrofurantoin and fosfomycin are only prescribed for UTI which excludes misclassification of the indication. Therefore, we believe that we at least could say something about the compliance of GPs to guidelines in the studied period. Therefore, we revised the aim in the abstract which reads as following:

Here, in this study, we describe antimicrobial drug prescribing patterns for UTIs by GPs and investigate adherence of prescribers in relation to the Dutch primary care guidelines in a large electronic primary care database.

and in the introduction:

This study aims to use a large GP database electronic primary care database to investigate the adherence of GPs to the Dutch UTI guidelines, by studying study the choice of antimicrobial drugs prescribed for UTIs by GPs and the duration of nitrofurantoin use over time in relation to the Dutch national guidelines for the treatment of UTIs in primary care.

Additionally, I would like to comment some other improvable points:

Abstract:

- explain the meaning of IPCI the first time it appears

Thank you for noting this, we have included this.

- clarify the difference between the two populations described (approximately 2.5 million patients and 1,755,085 patients)

The 2.5 million patients consists of the source population namely all patients for whom data is collected in the IPCI database. From the database, we then selected a study population based on inclusion and exclusion criteria which is described in the method section. We, however, made a selection, which is described in the method section. We selected only patients with at least one year of database follow-up. We further clarified this in the abstract.

It now reads as:

All patients ≥ 12 years with at least one year of follow-up from 1996 to 2014 were extracted from the database..

- indicate the guidelines versions used (in the results you mention the 2005 and 2013)

The guidelines for GPs have been changed several times in the period that we have studied. The first guidelines were from 1989. We have used especially the guidelines of 1999, 2005 and 2013 to study the adherence of the GPs. The guidelines were added as references (4-7) We added the guidelines to the abstract:

The choice of antimicrobial drugs classes for UTIs and the duration of nitrofurantoin use in women were compared to the Dutch primary care guidelines of 1989, 1999, 2005 and 2013.

- in the results: it is not clear if the increase in the proportion of prescriptions for UTI within all prescriptions with a code is due to a change in the number of prescriptions with a code or to a real increase in the number of prescriptions for UTIs.

Indeed, this is a point that we also considered. We see an increase of UTI prescriptions over the years. However, we are aware of the fact that better coding causes this increase. Therefore, we looked into the nitrofurantoin prescriptions. Nitrofurantoin is and was only prescribed for UTIs. We found that the nitrofurantoin prescriptions without a code flatten or even slightly decrease, whereas the coded nitrofurantoin prescriptions increased over time. Because this is still not conclusive. We calculated the number of antimicrobial drug prescriptions for UTIs as a proportion of all antimicrobial drug prescriptions with an indication code. Here, we also found an increase. We clarified this sentence:

Moreover, we calculated the proportion of antimicrobial drug prescriptions for UTI with as denominator all antimicrobial drug prescriptions (for all indications) with an indication code.

Moreover, be cautious in the interpretation of the results obtained (e.g. “with a significant effect of the guideline of 2005”)

For nitrofurantoin, there was a real increase in the number of prescriptions and this drug has no other indications than UTI. But we agree that we cannot prove that the effect is the only and direct effect of the guideline, we therefore changed the sentence that we found a difference between the period before and after implementation of the guideline as follows:

In women, UTIs were increasingly ($p < 0.05$) treated with nitrofurantoin derivatives with a statistically significant effect of the difference after implementation of the guideline of 2005.

Strengths and limitations:

please, explain better / improve the writing of the second and third points. They both result confusing

Thank you for noting this and giving us the opportunity to strengthen these points. We rewrote the second and third point:

- Every individual in the Netherlands is assigned to a single GP and antimicrobial drugs are not sold over the counter. This limits information bias, since we had access to all antimicrobial drug prescriptions prescribed in primary care.
- Improvement of coding discipline over calendar time in more recent years might bias conclusions on total prescribing of antimicrobial drugs, but we assume that this bias would be similar for all indications. Furthermore, we compared the antimicrobial drug prescription data for UTIs with antimicrobial drug prescription data for other infections.

Introduction:

instead of “a large GP database”, please use a more accurate terminology

Thank you for this suggestion. We adjusted this to: “a large electronic primary care database”.

Methods:

- The subsection “Study population” refers, additionally, to the type of study, data sources and variables studied. Please, indicate it.

Thank you for noting this. We subdivided this section into multiple sections to indicate this.

- What does the inclusion criteria “with at least one year of valid database history” mean? Could it have any effect on the study population selection?

Similar to all other studies which we conduct using the IPCI database, we only included patients for whom we had at least one year of valid database history. This strategy is taken to exclude non-residents (tourists or people living elsewhere who consult the GP only once) and to ensure that – for each patient – we have sufficient information on medical history prior to start of follow-up. Based on this inclusion criterium, we indeed did not include all patients present within the IPCI database but it is unlikely that this has biased the results.

- Which version of ICPC codes did you use? Detail the ICPC codes included in each indication group.

We have used the ICPC codes version 5 of 2013. We have added a supplementary table to indicate which codes were used. The table include all indication codes that we studied. The codes with asterisk indicate the codes that we assigned to the group of urinary tract infections.

- Analyses: form my point of view, presenting the results as number/proportion of prescriptions and users results confusing and difficult to understand. My recommendation is including just prescribing rates (proportion of users), as most studies do.

We agree that the proportion of users is the most frequently used measure in pharmacoepidemiologic studies. However, in light of antimicrobial resistance not only the number of users of antimicrobial drugs is interesting, but it is also very interesting to know the total number of antimicrobial drug prescriptions. The number of antimicrobial drugs prescribed to one specific individual varies largely. Therefore, we really would like to keep both figures. We explained this in the methods section:

Since both the number of users, but also the total number of antimicrobial drug prescriptions are interesting with regard to the study of antimicrobial resistance, we studied both.

- You mention the calculation of prevalence, but it is an incidence. Please, correct it.

We therefore changed the term into frequency of prescriptions:

Because of the dynamic nature of the study cohort, the annual frequency of antimicrobial drug prescriptions was calculated by dividing the total number of antimicrobial drug prescriptions by the total number of person-years (PY).

- The last paragraph in the Methodology section must be rewritten. It is not clear. Moreover, you refer to "calendar time" throughout the text. What do you exactly mean?

Thank you for this suggestion. We rewrote the last paragraph, resulting in the following:

Finally, we used a time series ARIMA model in SPSS 24 to determine the effect of the implementation of the guideline of 2005. The calendar year was first univariably added to the analysis to study differences of antimicrobial prescribing over time. Next, we ran a model including calendar year and the intervention (implementation of the guideline). When the p-value of the interaction term was < 0.05, the difference in slope before and after the implementation of the guideline was significant, implying a significant effect of the implementation of this guideline.

With calendar time, we actually just mean time thus 1996, 1997,2014. To avoid confusion with time since start of the study, we called this calendar time to distinguish it from follow-up time.

Results:

- I do not understand what the mean follow-up time means. Please clarify it according to the information provided in the Methods section.

The mean follow-up time is the mean time of follow-up of all patients included in the study.

We added the collection of the follow-up time of each patient in the database in the methods section:

Gender, age at time of prescription(s) and follow-up time (time of each patient in the database) were assessed for all patients.

- Only 56.7% of the prescriptions were linked to an indication. Discuss the possible bias

Since only 56.7% of the prescriptions were linked to an indication, we do not know the indication of the other 43.3% of antimicrobial drugs which were prescribed. It is unlikely however that coding or not-coding the indication of use would be driven by the underlying infection. It is more likely that

certain GPs (also depending on the year in which the antibiotics were prescribed; because coding increased in more recent years) didn't assign indication codes in general instead of only for certain indications. Coding discipline improved over time, possibly partly because of increased health insurance requirements.

- Ideas described in lines 158-161 and 167-173 refer to Methodology and should be included in the corresponding section.

We deleted it and added to the methods:

The prescribing of antimicrobial drugs by GPs was compared to the recommendations according to the national GP guidelines (Table 1).

Furthermore, we agree that the idea in line 167-173 is not clearly explained in the methods section, so we included there the following:

We intended to investigate if improved coding over time has influenced the results of the frequency of prescribing antimicrobial drug prescriptions for UTIs in general, independent of the drug class. Therefore, we also studied the frequency of nitrofurantoin prescriptions over time (since nitrofurantoin was only prescribed for UTIs) and we studied the proportion of prescriptions for UTIs within total prescriptions with an indication code (thus including all antimicrobial drug prescriptions with information on indication of use).

However, for the readability of this paragraph, we would like to keep this sentence in the results.

- please explain the meaning of AB acronym

Thank you for noting this. This should be antimicrobial drugs. We corrected this throughout the manuscript.

- the title of the subsection "Adherence to guidelines" should be modified according to the idea explained at the beginning of this comment (the objective set is not completely reached, because you do not study adherence to guidelines. You are exploring the possible effect of new recommendations on the choice and duration of treatment, but you cannot attribute the changes in prescribing pattern to changes in guidelines)

We changed the title into:

Choice of antimicrobial drug prescription in relation to the recommendation of the national primary care guidelines

Discussion:

First and second paragraphs contain repeated interpretations and ideas. Please, sum them up. Moreover, I consider that the results obtained should be discussed and interpreted more in depth. If it is lacking, then the manuscript results a bit simple and trivial since it consists on a simple descriptive analysis. Although more clinical data available would be needed in order to assess the adherence, an interesting discussion is behind these data so I encourage to improve its discussion.

Thank you for these suggestions to improve the quality of our discussion. We made some changes, which you can see in the text.

References:

There are some errors in bibliography, such as the lack of uniformity in references or the incorrect organizations name.

Thank you, we corrected this.

Figures:

- please, simplify the figure legends. They contain ideas already mentioned in the text.
- As I previously mentioned, it results confusing to include the users and prescription results
- Modify the colors in order to facilitate interpretation when printed in black and white.

We changed the legends and figures accordingly.

Supplementary table 1 needs a more descriptive title.

Thank you for this suggestion. We changed it into:

ATC codes of antimicrobial drugs prescribed by GPs to treat urinary tract infections

Reviewer: 2

In an era of increasing appreciation of the public health threat posed by antibiotic resistance and the need for antimicrobial stewardship, this study addresses an important and topical issue. The main conclusion highlighted by the authors is that databases with antimicrobial drug prescriptions and indications are useful for surveillance of antimicrobial drug prescribing over time. However, similar studies have previously been published in other countries that have established this principle and the main novelty of this study is thus restricted to the description of a surveillance system that can be used in the Netherlands.

Thank you for your suggestions. It is indeed true that this study confirms earlier studies about the usefulness of databases like IPCI to study surveillance of antimicrobial drug prescriptions over several years. However, we think our study is quite unique for its long study period.

The study throws up some potentially interesting findings in terms of adherence of prescribers to national guidelines and the changes over time. However, at this stage the findings are primarily descriptive with little insight into the underlying causes. This severely limits both understanding of what factors influence the behaviour of prescribers and options for developing potential interventions to improve stewardship and the quality of prescribing at the present time.

We thank the reviewer for this suggestion and added this limitation to the discussion. It would indeed be very interesting to understand the underlying reasons of choices of GPs when prescribing antimicrobial drugs. When we are aware of these choices, we could look for interventions to change "bad" choices and thus improve antibiotic stewardship, which is very important to prevent the development of antimicrobial resistance. Unfortunately, with the present data, we could not look any further in the behavior of GPs. We do think that further studies looking into reasons for deviating from the guidelines are important and we encourage further studies on this subject. We elaborated about this in the discussion:

Additionally, although we found that antimicrobial drugs were increasingly prescribed according to the guidelines, the study design did not allow us to investigate the reasons behind deviations from the guidelines. This is of interest and should be studied further eventually by qualitative prospective research to distinguish between forced deviations caused by resistant pathogens and other deviations, which should be diminished to promote good antimicrobial stewardship.

The authors are to be encouraged to continue their surveillance but further work is needed to give more insight into the factors influencing prescribing behaviour. For example, what is known of the variation in patterns of GP prescribing of antibiotics? Is there significant geographical/regional variation and if so does this correlate with other factors such as urban/rural settings or levels of deprivation. When new treatment guidelines are issued in the Netherlands, what is the process for disseminating this information to prescribers and might it result in variation in uptake by (e.g. are prescribers actively notified of changes or is the onus on them to try to keep up to date with guidance). Are data available as to the ages of GPs and how long they have been in practice? Might older GPs be more set in their ways and less amenable to changing their prescribing practice when new guidelines are issued? Additional information of this nature will serve to markedly improve the value and translational application of this potentially valuable work.

Thank you for these interesting thoughts. The NHG (Nederlands Huisartsen Genootschap (Dutch College of General Practitioners)) is the scientific society of GPs with the aim to promote scientific professional practice of GPs. They develop guidelines for diagnostics and treatment of disorders which are common in GP practice. The NHG organizes meetings in which GPs can discuss new guidelines. It is not compulsory for GPs in The Netherlands to become member of the College, however, currently more than 95% of all GPs are a member of the College. Members of the College are stimulated to prescribe according to the guidelines and new guidelines are actively distributed and discussed, amongst others during trainings.

As the reviewer suggested, it would be interesting to investigate whether differences in prescribing behavior could be attributed to regional differences, differences by age of the prescriber and/or differences by age of the GP practice, but unfortunately, because of confidentiality, we do not have access to these data. Based on literature however; we are able to say something about resistance patterns between the different regions. A study with data from 2009 did not see differences in susceptibility of E.coli's isolated from urinary tract infections of different regions, except for a significant difference in susceptibility to amoxicillin-clavulanic acid between the northern and eastern region of The Netherlands.²

These guidelines are actively distributed by the Dutch College of General Practitioners.

Unfortunately, because of confidentiality, we could not investigate differences in prescribing patterns by region nor by GP characteristics such as age and/or gender.

1. Tell D, Engstrom S, Molstad S. Adherence to guidelines on antibiotic treatment for respiratory tract infections in various categories of physicians: a retrospective cross-sectional study of data from electronic patient records. *BMJ Open* 2015;5(7):e008096.

2. den Heijer CD, Donker GA, Maes J, et al. Antibiotic susceptibility of unselected uropathogenic Escherichia coli from female Dutch general practice patients: a comparison of two surveys with a 5 year interval. *J Antimicrob Chemother* 2010;65(10):2128-33.

VERSION 2 – REVIEW

REVIEWER	Sara Malo University of Zaragoza Spain
REVIEW RETURNED	14-Feb-2019

GENERAL COMMENTS	All the suggested modifications have been addressed and the manuscript is suited to be accepted. As far as I am concerned, no more changes are proposed.
--